# NEURAL MARKOV LOGIC NETWORKS

## ABSTRACT

We introduce Neural Markov Logic Networks (NMLNs), a statistical relational learning system that borrows ideas from Markov logic. Like Markov Logic Networks (MLNs), NMLNs are an exponential-family model for modelling distributions over possible worlds, but unlike MLNs, they do not rely on explicitly specified first-order logic rules. Instead, NMLNs learn an implicit representation of such rules as a neural network that acts as a potential function on fragments of the relational structure. Interestingly, any MLN can be represented as an NMLN. Similarly to recently proposed Neural theorem provers (Rocktäschel & Riedel, 2017), NMLNs can exploit embeddings of constants but, unlike NTPs, NMLNs work well also in their absence. This is extremely important for predicting in settings other than the transductive one. We showcase the potential of NMLNs on knowledge-base completion tasks and on generation of molecular (graph) data.

## 1 INTRODUCTION

Parameters for a statistical relational model are typically estimated from one or more examples of relational structures that typically consist of a large number of ground facts. Examples of such structures are social networks (e.g. Facebook), protein-protein interaction networks, the Web, etc. A challenging task is to learn a probability distribution over such relational structures from one or few examples. One solution is based on the assumption that the relational structure has repeated regularities; this assumption is implicitly or explicitly used in most works on statistical relational learning. Then, statistics about these regularities can be computed for small substructures of the training examples and used to construct a distribution over the relational structures. Together with the maximum-entropy principle, this leads to distributions such as Markov logic networks (Richardson & Domingos, 2006; Kuželka et al., 2018)

In this paper, we propose Neural Markov Logic Networks (NMLN). Here, the statistics which are used to model the probability distribution are not known in advance, but are modelled as neural networks trained together with the probability distribution model. This is very powerful when compared to classical MLNs, where either domain experts are required to design some useful statistics about the domain of interest by hand (i.e. logical rules) or structure learning based on combinatorial search needs to be performed. These requirements normally limit a wide application of these models as out-of-the box tools. It is worth noticing that overtaking the need of such "feature-engineering" is one of the reasons behind the massive adoption of deep learning techniques. However, not much has been done in the same direction by the statistical relational learning community. Moreover, designing statistics as neural networks allows a more fine-grained description of the data, opening the doors to applications of our model to the generative setting.

### CONTRIBUTIONS

The main contributions of this work are: *(i)* we introduce a new statistical relational model, which overcomes actual limitations of both classical and recent related models such as (Richardson & Domingos, 2006; Rocktäschel & Riedel, 2017; Sourek et al., 2018); *(ii)* we propose a theoretical justification of the model as naturally emerging from a principle of Min-Max-entropy; *(iii)* we provide a Tensorflow implementation of this model; and *(iv)* we showcase its effectiveness on two quite diverse problems: knowledge-base completion and generative modelling of small molecules.

RELATED WORK

The need to extend relational models with neural components is a topic that has been receiving increasing attention in the last few years. An integration of logic reasoning and neural models was proposed based on fuzzy logic (Serafini & Garcez, 2016; Diligenti et al., 2017; Marra et al., 2019). Here, neural models implementing FOL relations are optimized in order to satisfy differentiable approximations of logical formulas obtained by means of fuzzy t-norm theory. However, the lack of probabilistic arguments allows a sound application of such fuzzy-logic based methods only to hard-constrained settings. In Manhaeve et al. (2018), the probabilistic logic programming language ProbLog (De Raedt et al., 2007) is extended to allow probabilities of atoms to be predicted by neural networks and to exploit differentiable algebraic generalizations of decision diagrams to train these networks. Lifted relational neural networks (Sourek et al., 2018) unfold neural networks with shared weights, as in convolutional networks, using forward chaining. A semantically very similar approach was the one in Rocktäschel & Riedel (2017), where the authors implemented the Prolog backward chaining with a soft unification scheme operating on constants and relations embeddings. The proposed Neural Theorem prover was able to exploit the geometry of the embedding space to improve its reasoning capabilities, but the need for embeddings prevents this model to be applicable to settings different from the transductive one. Most importantly, neither of these latter two works provides means for probabilistic modelling of relational structures.

The idea of exploiting neural networks to extract regularities in non-euclidean settings has been recently revisited by the deep learning community in the context of Graph Neural Networks (GNN) models (Scarselli et al., 2009; Defferrard et al., 2016; Xu et al., 2018). In GNNs, latent representations of nodes are obtained by an aggregation of neighboring nodes representation by means of an iterative diffusion mechanism. However, the inference is performed only on neighborhoods induced by the actual connections of the graph, preventing the exploitation of these models for modeling distributions of structural properties of the graph. Lippi & Frasconi (2009) was an early attempt to integrate MLNs with neural components. Here, an MLN was exploited to describe a conditional distribution over ground atoms, given some features of the constants. In particular, the MLN was reparametrized by a neural network evaluated on input features. However, this method still relied on hand-crafted logical rules for modelling the distribution.

## 2 PRELIMINARIES

This paper follows the setting of so-called Model A from Kuželka et al. (2018). We consider a function-free first-order logic language $\mathcal{L}$, which is built from a set of constants $\mathcal{C}$ and predicates $\mathcal{R} = \bigcup_i \mathcal{R}_i$, where $\mathcal{R}_i$ contains the predicates of arity i. For $c_1, c_2, \ldots, c_m \in \mathcal{C}$ and $R \in \mathcal{R}_m$, we call $R(c_1, c_2, \ldots, c_m)$ a *ground atom*. We define *possible world* $\omega \in \Omega$ as the pair $(C, A)$, where $C \subseteq \mathcal{C}$, $A$ is a subset of the set of all ground atoms that can be built from the constants in $C$ and any relation in $\mathcal{R}$. We define the size of a possible world $n = |C|$ and $\Omega$ is the set of all possible worlds. A *fragment* $\omega\langle S \rangle$ is defined as the restriction of $\omega$ to the constants in $S$. It is again a pair $\omega\langle S \rangle = (S, B)$, with $S$ the constants of the restriction and $B$ a set of ground atoms which only use constants from $S$. Given a fragment $\omega\langle S \rangle$ and $k = |S|$, we can anonymize it by mapping the constants in $S$ with a permutation $\widehat{S}$ of the integer set $\{1, 2, ..., k\}$. We call this an *anonymized fragment* $\gamma$. Suppose we have a given world $\widehat{\omega}$ of size $n$, we define $\Gamma_k(\widehat{\omega})$ the collection of all the anonymized fragments of width $k$ of $\widehat{\omega}$. It is easy to verify that $|\Gamma_k(\widehat{\omega})| = \binom{n}{k} k!$. The collection $\Gamma_k(\widehat{\omega})$ is a multiset, since, after anonymization, multiple fragments could be identical. An example of the process of anonymization and of the identification of structural patterns among anonymized fragments is shown in Figure 1.

## 3 NEURAL MARKOV LOGIC NETWORKS

### 3.1 INTUITION AND FORMULATION

Given a world $\widehat{\omega} \in \Omega$, we are interested in *models* of the probability $P_\omega$, for a generic $\omega \in \Omega$. To this end, we want to compute statistics on fragments of the given $\widehat{\omega}$ and exploit them to construct a distribution on (possibly larger and smaller) relational structures $\omega$. Let us define $\phi(\gamma)$ as a *fragment potential function*, which is simply a function from anonymized fragments of width $k$

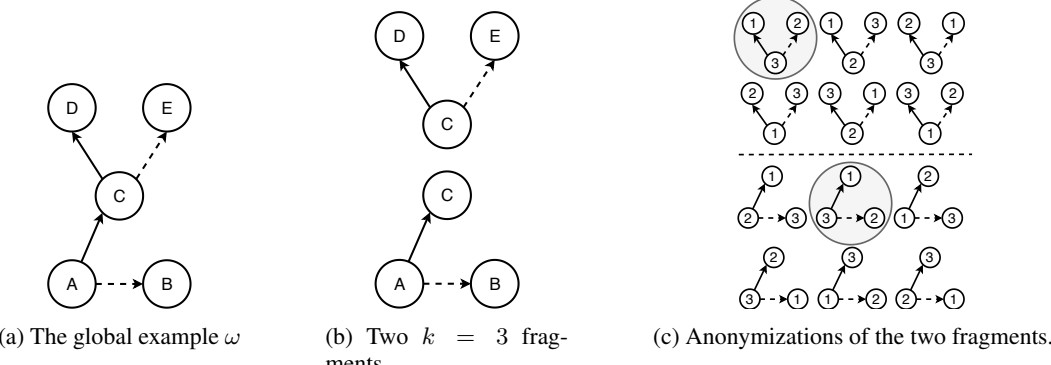

(a) The global example $\omega$      (b) Two $k = 3$ frag-
ments.      (c) Anonymizations of the two fragments.

Figure 1: **The process of individuating structural patterns in anonymized fragments.** White circles represent constants, while the two relations are represented as solid and dashed arrows (absence of an arrow means that the relation is false). The given world is shown on the left. Two possible fragments are shown in the middle. All their possible anonymizations are shown on the right. *Grey circles* show two repeated anonymized fragments found in two different fragments. The model exploits these regularities on fragments to model the distribution of the larger structure.

to real numbers. We can search for a maximum-entropy distribution $p(\omega)$ such that the following two expected values are the same: (i) the expected value of $\phi(\gamma)$ where $\gamma$ is sampled uniformly from $\Gamma_k(\widehat{\omega})$ and (ii) the expected value of $\phi(\gamma')$ where $\gamma'$ is sampled uniformly from $\Gamma_k(\omega)$ and $\omega$ is sampled, in turn, from $p(\omega)$. The intuition here is that, at least on average, the fragments of the given training example should look similar to the fragments of possible worlds sampled from the distribution. It follows from (Kuželka et al., 2018) that the resulting maximum-entropy distribution is an exponential-family distribution resembling Markov logic networks.

The max-entropy setting provides us with a sound starting point for designing statistical relational models that generalize classical models such as Markov random fields. However, a necessary condition for these models to be designed is that one can provide a set of statistics $\phi(\omega)$ describing the data. In this section, we show how to get rid of the need to provide statistics in advance and how to learn these statistics together with the probabilistic model in a differentiable manner.

Let us consider a *fragment neural potential function* $\phi(\gamma; \mathbf{w})$. It is a parametric function with parameters $\mathbf{w}$. Let $\Phi(\omega; \mathbf{w}) = \frac{1}{|\Gamma_k(\omega)|} \sum_{\gamma \in \Gamma_k(\omega)} \phi(\gamma; \mathbf{w})$ be the corresponding *global neural potential*. We need a learning principle which would allow us to find a good $\phi(\gamma; \mathbf{w})$ to describe our data. To this end, suppose we need to solve the maximum-entropy optimization problem, but without any constraint on the statistics. The maximum-entropy solution in this case is the uniform distribution, which assigns equal probability to all possble worlds $\omega$. Now, suppose we add a single constraint on a potential $\tilde{\Phi}(\omega)$. If this potential is informative and makes some worlds more likely than others, then the solution moves from the uniform distribution to another distribution with *lower* entropy. Using this intuition, we can have a scheme where we maximize entropy by selecting the maximum-entropy distribution and minimize it at the same time by choosing the most informative statistics.

The above considerations give rise to a *Min-Max-Entropy* model for the target probability distribution $P_\omega$, which we call *Neural Markov Logic Network* and which we describe in turn. Let us first define the Max-Entropy problem with the new neural potentials (stated here as minimization of negative entropy):

$$\min_{P_\omega} \quad \sum_\omega P_\omega \log P_\omega \tag{1}$$

$$\text{subject to} \quad \sum_\omega P_\omega = 1, \ \forall \omega \colon p_\omega \geq 0 \tag{2}$$

$$\forall i \colon \mathbb{E}_{P_\omega}[\Phi_i(\omega; \mathbf{w}_i)] = \Phi_i(\widehat{\omega}; \mathbf{w}_i) \qquad \text{with } 0 < i \leq M \tag{3}$$

For fixed $\mathbf{w}_i$'s, we can use Lagrangian duality to obtain the following solution of the maximum entropy problem: $P_\omega = \frac{1}{Z} \exp\left(\sum_i \beta_i \Phi(\omega; \mathbf{w}_i)\right)$. Here, $Z$ is a normalization constant and the parameters $\beta_i$ are solutions of the dual problem $\max_{\beta_i} \left\{\sum_{i=1}^{M} \beta_i \Phi_i(\widehat{\omega}; \mathbf{w}_i) - \log Z\right\}$, which coincides with maximum-likelihood.[1]

Next we still need to incorporate the minimization of entropy by optimizing $\mathbf{w}_i$'s. Let us denote by $H(\beta_1, \ldots, \beta_M, \mathbf{w}_1, \ldots, \mathbf{w}_M)$ the entropy of the distribution $P_\omega$. Now, as previously introduced, the selection of the optimal values $\mathbf{w}_i$ is governed by the principle of minimization of entropy, leading to the optimization problem: $\min_{\mathbf{w}_i} \max_{\beta_i} H(\beta_1, \ldots, \beta_M, \mathbf{w}_1, \ldots, \mathbf{w}_M) = -\max_{\mathbf{w}_i} \min_{\beta_i} -H(\beta_1, \ldots, \beta_M, \mathbf{w}_1, \ldots, \mathbf{w}_M)$ subject to the constraints (2) and (3). Plugging in the dual problem and using strong duality, we obtain the following unconstrained optimization problem which is equivalent to the maximization of log-likelihood: $\max_{\mathbf{w}_i, \beta_i} \left\{\sum_{i=1}^{M} \beta_i \Phi_i(\widehat{\omega}; \mathbf{w}_i) - \log Z\right\}$. The maximization of the log-likelihood will be carried out by a gradient-based optimization scheme. The gradients of the log-likelihood w.r.t. to both the parameters $w_{i,j}$, where $w_{i,j}$ denotes the $j$-th component of $\mathbf{w}_i$, and $\beta_i$ are:

$$\frac{\partial \log(P_{\widehat{\omega}})}{\partial w_{i,j}} = \beta_i \left(\frac{\partial \Phi_i(\widehat{\omega}; \mathbf{w}_i)}{\partial w_{i,j}} - \mathbb{E}_{\omega \sim P}\left[\frac{\partial \Phi(\omega; \mathbf{w}_i)}{\partial w_{i,j}}\right]\right) \tag{4}$$

$$\frac{\partial \log(P_{\widehat{\omega}})}{\partial \beta_i} = \left(\Phi_i(\widehat{\omega}; w_i) - \mathbb{E}_{\omega \sim P}[\Phi_i(\omega; w_i)]\right) \tag{5}$$

At a stationary point, Eq. 5 recovers the initial constraint on statistics imposed in the maximization of the entropy. However, the minimization of the entropy is mapped to a new requirement: at stationary conditions, the expected value of the gradients of the $\Phi_i$ under the distribution must match the gradients of the $\Phi_i$ evaluated at the data points.

## 3.2 VECTOR EMBEDDINGS OF DOMAIN ELEMENTS

By anonymizing a fragment, the model loses any trace of the identity of the constants involved in it, preserving only their structural behaviours. While this feature is essential to allow the identification of structural patterns also inside a single possible world, it prevents the model from having different behaviour on specific constants. This, instead, is a basic feature of many existing transductive models, like NTP (Rocktäschel & Riedel, 2017), which exploit the geometry of a latent representation space of constants to improve their prediction capabilities.

To this end, we define an *embedding fragment neural potential* $\phi_e(\gamma, \widehat{S}; \mathbf{w}, \Theta)$, which is function of the anonymized fragment but also of the specific constants involved in it (i.e. the list of constants $\widehat{S}$). In particular, in transductive settings, we always have a possible world $\widehat{\omega}$ and we use the same constant set $S$ both during learning and inference. Let $\Theta \in \mathbb{R}^{n \times d}$ be a variable embedding matrix. It can be considered a map from the constant set $S$ to a latent real domain $\mathbb{R}^d$, i.e. the embedding space. Let $c(\widehat{S}, \Theta)$ be a function that concatenates the $k$ rows of $\Theta$ corresponding to the $k$ constants in the restricted set $\widehat{S}$. Thus, the embedding fragment neural potential $\phi_e$ can be seen as a function of both $\gamma$, which encodes the structural properties of the fragment and $c(\widehat{S}, \Theta)$, which encodes the identity of constants by providing a latent representation for them. In other words, $\phi_e(\gamma, \widehat{S}; \mathbf{w}, \Theta) = f(\gamma, c(\widehat{S}, \Theta); \mathbf{w})$ for some neural function $f$ parameterized by $\mathbf{w}$. This is inspired by works in the NLP community (Mikolov et al., 2013; Mnih & Kavukcuoglu, 2013), where the $c$ function can have different forms than concatenation. The components of the embedding vectors are treated as any other weights of the potential functions and are updated using gradients computed according to Eq. 4. Intuitively, the contrastive nature of the learning (Bordes et al., 2013; Trouillon et al., 2017),

---

[1]We note that the derivation of the dual problem follows easily from the derivations in (Kuželka et al., 2018), which in turn rely on standard convex programming derivations from (Boyd & Vandenberghe, 2004; Wainwright et al., 2008). Throughout this section we assume that a positive solution exists, which is needed for the strong duality to hold; this is later guaranteed by adding noise during learning.

leads to the development of similar embeddings for similar constants. As we show in Section 4.2, the addition of embedding of constants helps improving the prediction capability of our model in transductive settings.

### 3.3 INFERENCE

In order to design an optimization procedure to learn Neural Markov Logic Networks, we need to rely on some methods to sample from the distribution. In this paper, we exploit MCMC methods, in particular approximate Gibbs Sampling (GS) (Robert & Casella, 2013), to sample from Neural Markov Logic Networks. The approximation comes from the fact that GS requires a large number of steps before converging to the target distribution. However, we run it only for a limited number of steps $t$, which, in some cases, is restricted to $t = 1$. When this happens, our method recovers a discrete version of the Contrastive Divergence (CD) algorithm (Hinton, 2002).

Gibbs sampling cannot effectively handle distributions with a lot of determinism. In normal Markov logic networks, sampling from such distributions may be tackled by an algorithm called MC-SAT (Poon & Domingos, 2006). However, MC-SAT requires an explicit logical encoding of the deterministic constraints, which is not available in Neural Markov Logic Networks where deterministic constraints are implicitly encoded by the potential functions. In fact, only constraints that are almost deterministic, i.e. having very large weights, can occur in Neural Markov Logic Networks but, at least for Gibbs sampling, the effect is the same. Such distributions would naturally be learned in our framework on most datasets. Our solution in this paper is to simply avoid learning distributions with determinism by adding noise during training. In particular, we set a parameter $\pi_n \in [0, 1]$ and, at the beginning of each training epoch, we inverted each ground atom of the input possible worlds ($True$ to $False$ and vice versa) with probability $\pi_n$. Moreover, this added noise prevents the model to perfectly fit training data, acting as a regularizer (Bishop, 1995).

## 4 EXPERIMENTS

### 4.1 IMPLEMENTATION DETAILS

We implemented Neural Markov Logic Networks in Tensorflow. In order to maximally exploit the parallel computations capabilities of GPUs, multiple Markov chains are run in parallel. This is also useful because expected values of gradients (see Eq. 4 and 5) are computed on uncorrelated samples, while sequential samples sampled from a unique chain are known to be highly correlated.

In experiments, the different global neural potentials $\Phi_i$ can rely on fragments of different sizes $k$ so that for small $k$, the model can focus on learning very local statistics of the data, while, for large $k$, the model can focus on learning statistics on larger substructures. For example, if we represent molecules as a relational structure (see Section 4.3), rings are inherently global statistics which cannot be captured by local properties. This example underlines the importance of the choice of $k$ for a correct modeling of the data distribution. However, since a single evaluation of $\Phi_i(w)$ requires a summation over $d = \binom{n}{k} k!$ number of terms, the number of elements of the sum grows exponentially with $k$ (and polynomially, but very fast, with $n$). So exploiting large $k$ is usually admissible only for small domain sizes $n$.

### 4.2 KNOWLEDGE BASE COMPLETION

In Knowledge Base Completion (KBC), we are provided with an incomplete Knowledge Base (KB) and asked to complete the missing part.

The KBC task is inherently in the transductive setting, since all the constants are exploited both during the training and testing phase. Moreover, data are provided in a positive-only fashion: we only know what is true and we cannot distinguish between unknown and false facts. Kuželka & Davis (2019) studied KBC tasks under the missing-completely-at-random assumption and showed consistency of learning by maximum-likelihood where both missing and false facts are treated in the same way as *false*. Hence, here we also provide both unknown and false facts as false facts during the training procedure.

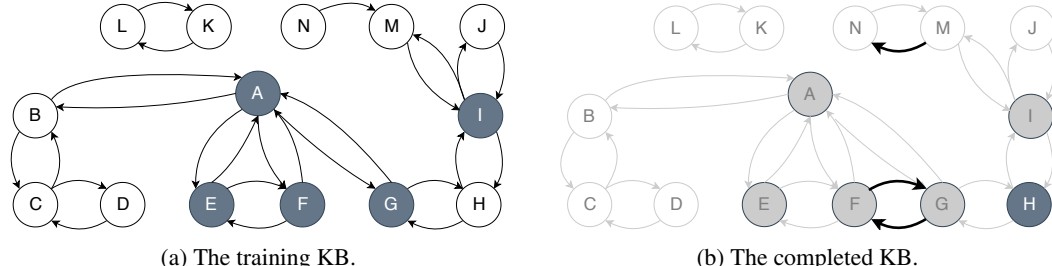

(a) The training KB.                    (b) The completed KB.

Figure 2: **Knowledge Base Completion in the Smokers dataset.** Circles represent constants. A grey circle means that the predicate `smokes` is *True*. A white circle means that the value of the predicate `smokes` is unknown. Links represent the relation `friendOf` (absence of an arrow means that the relation is *False*). The given world is shown on the top (2a), while the completed knowledge base is shown on the bottom (2b). The system learnt the symmetric nature of the friendship relation. It learnt that a friend of at least two smokers is also a smoker, and that two smokers, who are friends of the same person, are also friends.

**Smokers.** The "Smokers" dataset (Richardson & Domingos, 2006) is a classical example in statistical relational learning literature. Here, two relations are defined on a set of constants representing people: the unary predicate *Smokes* identifies those people who smoke, while the binary predicate *friendOf* maps people to their friend. This dataset is often used to show how a statistical relational learning algorithm can model a distribution by finding a correlation of smoking habits of friends. For example, in MLNs, one typically uses weighted logical rules such as: $\forall x \, \forall y \, \texttt{friendOf}(x,y) \rightarrow \texttt{smokes}(x) \leftrightarrow \texttt{smokes}(y)$. We learned a NMLN on the small smokers dataset. Since no prior knowledge about the type of rules that are relevant was used by NMLNs, the model itself had to identify which statistics are mostly informative of the provided data by learning the neural potential functions.

Here we use the Smokers dataset to define a Knowledge Base Completion task and to provide some basic intuitions about what kind of rules the model could have learned. In Figure 2, we show the setting before and after completion. In Figure 2b, we highlight only new facts whose marginal probability after training is significantly higher than the others, even though other facts have probabilities higher than the prior.

**Nations.** The Nations dataset (Kok & Domingos, 2007) provides information about properties and relations among countries as ground facts, like `economicaid(usa,israel)` or `embassy(israel,poland)`. There are $n = 14$ constants (i.e. nations), 56 relations and 2565 true facts. This dataset has been recently exploited for a KBC task by Rocktäschel & Riedel (2017), where some facts were removed from the dataset and the task was to predict them. The authors compared the performances of the state-of-the-art ComplEx neural model (Trouillon et al., 2017) with their proposed differentiable end-to-end neural theorem prover, showing that the combination of the two was able to outperform both of the models. Unary predicates were removed since the ComplEx model cannot deal with them. In this section, we show how we can use NMLNs to tackle a KBC task on the Nations dataset.

We implemented the fragment neural potentials $\phi(\gamma)$ as 2 hidden-layer neural networks, with sigmoidal hidden activations and linear output layer. The selection of the hyperparameters and the early-stopping epoch have been selected by means of a held-out validation set (the splits are same as the ones in Rocktäschel & Riedel (2017)). The size of layers has been selected from the interval $[75, 100, 150]$ for the first layer and $[30, 50, 100]$ for the second layer. The embedding size has been selected from the interval $[2, 3, 5, 10]$. The noise probability $\pi_n$ has been selected from the interval $[0, 0.01, 0.02, 0.03]$. The number of parallel chains has been selected from the interval $[10, 20, 30]$.

We followed the evaluation procedure in Rocktäschel & Riedel (2017). In particular, we took a test fact and corrupted its first and second argument in all possible ways such that the corrupted fact is not in the original KB. Subsequently, we predicted a ranking of every test fact and its corruptions to calculate MRR and HITS@m. The ranking is based on marginal probabilities estimated by running Gibbs sampling on the Neural Markov Logic Network; while training the network, we also run a

Table 1: MRR and HITS@$m$ on Nations.

| Metric | Model | | | | |
|--------|---------|------|---------|------|----------|
| | **ComplEx** | **NTP** | **NTP$\lambda$** | **NMLN** | **NMLN**-Emb |
| MRR | 0.75 | 0.75 | 0.74 | 0.77 | **0.81** |
| HITS@1 | 0.62 | 0.62 | 0.59 | 0.64 | **0.71** |
| HITS@3 | 0.84 | 0.86 | **0.89** | 0.86 | **0.89** |
| HITS@10 | **0.99** | **0.99** | **0.99** | **0.99** | **0.99** |

parallel Gibbs sampling chain on a state in which we fix the known part of the KB as true. Here, we compare the *ComplEx* model, the plain Neural Theorem Prover (*NTP*), the composition of the previous two (*NTP$\lambda$*), our plain model (*NMLN*) and our model when using potentials with embeddings (*NMLN-Emb*). In Table 1 we report the results of the KBC task on Nations. Both our models outperform competitors on the HITS@1 metric, with *NMLN-Emb* having a large gap over all the other models. It is interesting to note that the plain *NMLN* still performs better than differentiable provers, even if it is the only model which cannot exploit embeddings to perform reasoning and that has to rely only on the relational structure of fragments to make predictions. Finally, NMLN-Emb performs equally to or better than all the competitors in all the other metrics.

### 4.3 GRAPH GENERATION

One of the main features differentiating our model from standard MLNs is that we learn the statistics $\phi(\gamma)$ in a differentiable manner. The obtained probability distribution is then often far more fine grained than using predefined or hand-made statistics, that are limited to what the user considers important and do not search for other interesting regularities in the data. This opens the doors to the application of NMLNs to generative tasks in non-euclidean settings, which are receiving an increasing interest recently (You et al., 2018; Li et al., 2018).

In generation tasks, our model is asked to learn the probability distribution of the relational structures induced by a graph. Indeed, any FOL-description can be considered a multi-hyper graph; thus generating in the FOL setting is applicable to generating in any graph domain. In particular, to generate graphs, we can just use the same sampling technique used during training (i.e. Gibbs Sampling) to extract new samples.

In this section, we describe a molecule generation task. We used as training data the ChEMBL molecule database (Gaulton et al., 2016). We restricted the dataset to molecules with 8 heavy atoms. We used the RDKit framework [2] to get a FOL representation of the molecules from their SMILES encoding. In particular, we exploited only molecules having the most frequent atom types, i.e. `C`, `N`, `O`, `S`, `Cl`, `F`, `P`, and we encoded only two kinds of bonds: `SINGLE` and `DOUBLE`. A more detailed description of the data format is shown in the appendix.

We implemented the fragment neural potentials $\phi(\gamma)$ as neural networks with sigmoidal hidden activations and linear output layer. The hyperparameters were selected from the following ranges: the number of layers in $[1, 2]$; the hidden sizes of the layers in $[30, 100, 150, 200]$; the number of fragment potentials in $[1, 2]$, the size $k$ of potentials in $[2, 3, 4, 5, 6]$. The number of parallel chains was set to 5.

To qualitatively evaluate the results of this generative experiment, we follow Li et al. (2018), who designed an LSTM-based architecture for generative molecule modelling and applied it in a similar setting to ours. In Figure 3, we show a comparison between a sample of training data and a (random) sample of molecules generated by the proposed model. In particular, 20 generated samples are chosen randomly from the last 1000 samples extracted during the training procedure. By choosing them randomly, we avoided to have very correlated samples, which is inherent in the Gibbs sampling procedure. The generated samples resembles training data both in structural patterns and variety fairly well. Furthermore, in Figure 4, we compare the statistics, used in Li et al. (2018) for a similar task, on a sample of 1000 training and generated molecules. These statistics represent both general structural properties applicable to any graph as well as chemical structural properties of molecules

---

[2]https://rdkit.org/

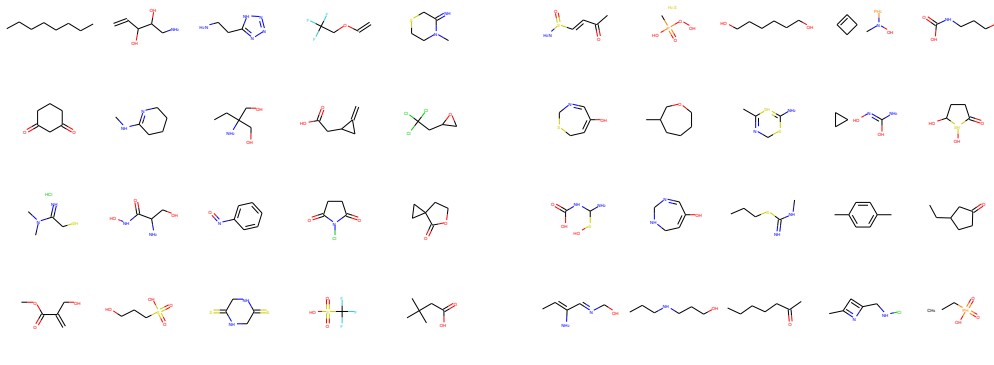

(a) Molecules from the training data.        (b) Generated molecules.

Figure 3: **Molecules generation.** A comparison between a sample of training data and a (random) sample of molecules generated by the proposed model. The generated samples fairly resembles training data both in structural patterns and variety. Better viewed in color.

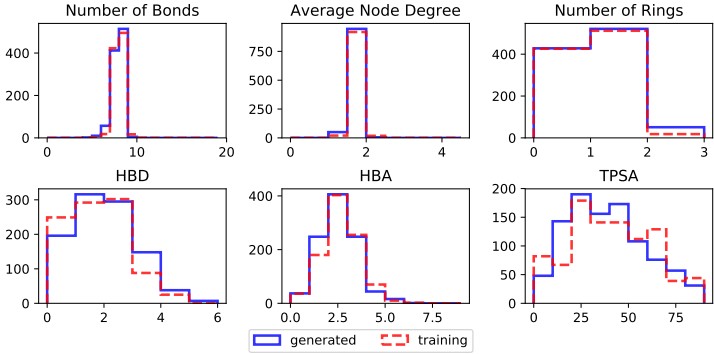

Figure 4: **Molecules generation.** Comparing the distributions of some chemical properties of the training data with the ones of generated data. The generated samples are capable of perfectly fitting structural properties and to very well resembling functional properties.

(e.g. the topological polar surface area (TPSA) is a topological indicator of the capability of a molecule to permeate a membrane as a function of the number of polar atoms it contains). These statistics were computed using the RDkit framework.

## 5 CONCLUSIONS

In this paper we have introduced Neural Markov Logic Networks, a statistical relational learning model combining representation learning power of neural networks with principled handling of uncertainty in the maximum-entropy framework. The proposed system works remarkably well on small domains. Although not explained in detail in this paper, it is also straightforward to add standard logical features as used in MLNs to NMLNs.

The main future challenge is making NMLNs scale to larger domains. At the moment NMLNs do not scale to large knowledge bases, which is not that surprising given that NMLNs can theoretically represent any distribution. A more work should therefore be done in the direction of identifying more tractable subclasses of NMLNs and exploiting insights from lifted inference literature (Braz et al., 2005; Gogate & Domingos, 2011; den Broeck et al., 2011).

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

# A   TRANSLATING MARKOV LOGIC NETWORKS TO NEURAL MARKOV LOGIC NETWORKS

Markov logic networks are a popular statistical relational framework. It turns out that every Markov logic network can be represented as a Neural Markov Logic Network with a *single* carefully selected potential function. We give details of the translation between these two frameworks below. Essentially what we need to show is that Model B from (Kuželka et al., 2018) can be translated to Model A from that same work, which is close enough to Neural Markov Logic Networks and so we can then easily encode the result to a Neural Markov Logic Network.

Kuželka et al. (2018) studies two maximum-entropy models, Model A, which is close to the model that we study in this paper, and Model B, which is the same as Markov logic networks. Syntactically, both models are encoded as sets of weighted first order logic formulas, e.g. $\Phi = \{(\alpha_1, w_1), \dots, (\alpha_M, w_m)\}$. In particular, given a positive integer $k$, Model A defines the following distribution:

$$p_A(\omega) = \frac{1}{Z} \exp \left( \sum_{(\alpha, w) \in \Phi_A} w \cdot \#_k(\alpha, \omega) \right)$$

where $Z$ is the normalization constant and $\#(\alpha, \omega)$ is the fraction of size-$k$ subsets $\mathcal{S}$ of constants in the possible world $\omega$ for which $\omega \langle \mathcal{S} \rangle \models \alpha$ (i.e. the formula $\alpha$ is classically true in the fragment of $\omega$ induced by $\mathcal{S}$). Let us first define

$$\phi_{\alpha, w}(\gamma) = \begin{cases} w & \gamma \models \alpha \\ 0 & \gamma \not\models \alpha \end{cases}$$

It is then easy to see that the distribution $p_A(\omega)$ can also easily be encoded as a Neural Markov Logic Network by selecting the potential function $\phi(\gamma) = \sum_{(\alpha, w) \in \Phi_A} \phi_{\alpha, w}(\gamma)$ and by carefully selecting the weights $\beta_i$ in the Neural Markov Logic Network.

Next we show that all distributions in Model B can be translated to distributions in Model A. First we will assume that the formulas $\alpha_i$ do not contain any constants.

Model B is given by

$$p_B(\omega) = \frac{1}{Z} \exp \left( \sum_{(\beta, v) \in \Phi_B} v \cdot n(\beta, \omega) \right)$$

where $n(\beta, \omega)$ is the number[3] of true injective groundings of the formula $\beta$ in the possible world $\omega$. Hence, Model B is exactly the same as Markov logic networks up to the requirement on injectivity of the groundings. However, as shown in (Buchman & Poole, 2015), any Markov logic network can be translated into such modified Markov logic network with the injectivity requirement on the groundings.

Let $k$ be an integer greater or equal to the number of variables in any formula in $\Phi_B$. Now, let $\Gamma$ be the set of all size-$k$ fragments. For every formula $\beta$ in $\Phi_B$, we introduce a partition $\mathcal{P}$ on $\Gamma$ induced by the equivalence relation $\sim_\beta$ defined by: $\gamma \sim_\beta \gamma'$ iff $n(\beta, \gamma) = n(\beta, \gamma')$. Since $\beta$ is assumed to not contain any constants, we can capture each of these equivalence classes $C$ by a (possibly quite big) first-order logic sentence without constants $\beta_C$. Let $C_i$ be the equivalence class that contains fragments $\gamma$ such that $n(\beta, \gamma) = i$. Let $m(\beta, \omega) = \sum_{C_i \in \mathcal{P}} \sum_{\gamma \in \Gamma_k(\omega)} i \cdot \mathbb{1}(\gamma \models \beta_C)$. By construction, it holds $m(\beta, \omega) = \sum_{\gamma \in \Gamma_k(\omega)} n(\beta, \gamma)$. Every true injective grounding of the formula $\beta$, having $l$ variables, is contained in $\binom{n-l}{k-l}$ different size-$k$ fragments of $\omega$, each of which gives rise to $k!$ anonymized fragments in the multi-set $\Gamma_k(\omega)$. So $m(\beta, \omega)$ is over-counting the number of true groundings $n(\beta, \omega)$ by a constant factor. It follows that, by carefully selecting the weights of the formulas $\beta_C$ we can encode the distribution $p_B(\omega)$ also in Model A. Although this particular transformation that we have just sketched is not very efficient, it does show that Neural

---

[3]In (Kuželka et al., 2018), Model B is defined using *fractions* of true grounding substitutions instead of *numbers* of true grounding substitutions. However, these two definitions are equivalent up to normalizations and both work for our purposes but the latter one is a bit more convenient here. Hence we choose the latter one here.

Markov Logic Networks with potential functions of width $k$ can express all distributions that can be expressed by Markov logic networks containing formulas with at most $k$ variables.

First-order logic formulas defining Markov logic networks may also contain constants. In Neural Markov Logic Networks we may represent constants using vector-space embeddings as described in the main text. One can then easily extend the argument sketched above to the case covering Markov logic networks with constants.

## B  ALGORITHMS

In the following, we will show a learning algorithm for NMLNs that relies on approximated Gibbs sampling.

The general learning algorithm for NMLN is described in Algorithm 1.

---
**Algorithm 1** NMLN general learning algorithm
---
**Input:** $\widehat{\omega}$: the given training world

 1: **procedure** LEARN($\widehat{\omega}$)
 2:     $\eta$: learning rate
 3:     $\tilde{\omega}$: $M$ randomly initialized Markov chain states
 4:     **while** stopping criterion **do**
 5:         **while** chains convergence criterion **do**
 6:             $\tilde{\omega} \leftarrow$ SAMPLE-STEP($\tilde{\omega}, M, \beta_i, w_i$)
 7:             $\frac{\partial \log(P_{\widehat{\omega}})}{\partial w_i} \leftarrow \beta_i \left( \frac{\partial \Phi_i(\widehat{\omega};w_i)}{\partial w_i} - \frac{1}{M} \sum_{\tilde{\omega}} \frac{\partial \Phi(\tilde{\omega};w_i)}{\partial w_i} \right)$
 8:             $\frac{\partial \log(P_{\widehat{\omega}})}{\partial \beta_i} \leftarrow \left( \Phi_i(\widehat{\omega};w_i) - \frac{1}{M} \sum_{\tilde{\omega}} \Phi_i(\omega;w_i) \right)$
 9:             $w_i \leftarrow w_i + \eta \frac{\partial \log(P_{\widehat{\omega}})}{\partial w_i}$
10:             $\beta_i \leftarrow \beta_i + \eta \frac{\partial \log(P_{\widehat{\omega}})}{\partial \beta_i}$
---

A possible sampling procedure, implementing the generic SAMPLE-STEP and exploiting Gibbs Sampling, is described in Algorithm 2

---
**Algorithm 2** Sampling Procedure
---
**Input:** $\tilde{\omega}$: the current states of the chains
**Input:** $M$: the number of chains
**Input:** $\beta_i, w_i$: current parameters

 1: **procedure** SAMPLE-STEP($\tilde{\omega}, M, \beta_i, w_i$)
 2:     $s \leftarrow 1$ *# sample index*
 3:     $i \leftarrow 1$ *# ground atom index*
 4:     $n$ number of ground atoms in $\omega$
 5:     **while** $s \leq M$ **do**
 6:         $\omega = \tilde{\omega}_s$ *# s-th chain*
 7:         **while** $i \leq n$ **do**
 8:             $p \leftarrow P_{\omega_i=1|\omega_{j \setminus i}}$
 9:             $r \in [0,1]$ from a uniform distribution
10:             **if** $r < p$ **then**
11:                 $\omega \leftarrow \omega_i = 1$
12:             **else**
13:                 $\omega \leftarrow \omega_i = 0$
14:             $i \leftarrow i + 1$
15:         $\tilde{\omega}_s = \omega$
16:         $s \leftarrow s + 1$
17:     **return** $\tilde{\omega}$
---

Figure 5: An example of molecule

## C  GENERATING MOLECULES

Even though molecules can be described with a high level of precision, using both spatial features (i.e. atoms distances, bond length etc.) and chemical features (i.e. atom charge, atom mass, hybridization), in this work, we focused mainly on structural symbolic descriptions of molecules.

In particular, we described a molecule using two sets of FOL predicates:

- *Atom-type unary predicates*: these are `C`, `N`, `O`, `S`, `Cl`, `F`, `P`.
- *Bond-type binary predicate*: these are `SINGLE` and `DOUBLE`.

An example of a molecule FOL description can be:

```
O(0), C(1), C(2), C(3), N(4), C(5), C(6), C(7), O(8), O(9)
SINGLE(0,1), SINGLE(1,0), SINGLE(1,2), SINGLE(2,1), SINGLE(2,3)
SINGLE(3,2), SINGLE(3,4), SINGLE(4,3), SINGLE(4,5), SINGLE(5,4)
SINGLE(5,6), SINGLE(6,5), SINGLE(5,7), SINGLE(7,5), DOUBLE(7,8)
DOUBLE(8,7), SINGLE(7,9), SINGLE(9,7), SINGLE(6,1), SINGLE(1,6)
```

