# OpenReview forum: "Neural Markov Logic Networks"
_ICLR.cc/2020/Conference — Reject_

### Official Review · AnonReviewer2 · 2019-10-23
**Official Blind Review #2**

**Rating:** 1

**Review:**

The following paper provides an extension to Markov Logic Networks(MLNs), by removing their dependency on pre-defined first-order logic rules.  This is handled via neural networks which are able to capture the statistical relations, so-called Neural Markov Logic Networks(NMLNs). As this is an implicit representation from the neural network, the rules act as potential functions on the MLN structure. As general MLN techniques are reliant on domain experts or exhaustive structure learning approaches, NMLNs are able to model more domains as provided in the work with knowledge-base completion tasks and generative modelling of molecules.

The primary contribution in this body of work is based on the observation that relational structure repeats regularities in the data, and where deriving the statistics of these regularities is what allows for improved accuracy in a model.  The proposed NMLN  is architecturally identical to MLNs with the difference being the addition of the potential function.

As defined by the paper, fragments are connected subsets derived from relational data. The authors derived sets of fragments with constants defined by values in the data and anonymized fragment sets with integer assignments. With potential functions sampled from the anonymized and the true value fragments. The objective is a search for a maximum-entropy distribution to model the data derived fragments. The neural network aspect comes in the form of the minimum-maximum entropy modelling with weights for given fragments being learned by minimising the entropy of the fragment potential function. Where the model also maximizes the log-likelihood related to the anonymized fragments. The intuition in this work is that by selecting the maximum entropy distribution while also minimizing it by selecting the most informative statistical information for it, we will derive an accurate probability distribution given the possible worlds.
Overall, the paper performs a well enough job explaining the technical aspects. It does a thorough job explaining the algorithmic detail in the main body, and the appendix provides clear and implementable pseudocode equations. It is not exactly clear to me why the anonymization of fragments is necessary, but the authors suggest this places a greater focus on the graph structure and minimizes the model acting differently with different constants. The min-max entropy modelling also appears to be a novel approach in terms of statistical relational modelling. The results also demonstrate the success of NMLNs modelling on relational data and KB completion.

Regarding the technical aspects, a few concerns are the claims that the domains for experimentation seem rather trivial since the smoking and nations dataset are common relational datasets, and the strength of this model is the ability to learn on other domains. This is possibly addressed with the molecule experiments, but more datasets would have helped in confirming the breadth of domains as claimed by the paper. It is also difficult to measure the success of their model with generative modelling as no baseline was present for the molecule experiments. I still view this paper as a positive contribution to MLN research, with a technique that is successful among the few experiments tested.

At the same time, however, given the emerging contributions in the area, such as relational neural networks (Poole et al.), although the technical development is sensible, I don’t find the contribution that technically novel, and seems somewhat straightforward. Given the preliminary investigations, this is a reject from my side.

**Experience Assessment:**

I have published one or two papers in this area.

**Review Assessment: Checking Correctness Of Derivations And Theory:**

I assessed the sensibility of the derivations and theory.

**Review Assessment: Checking Correctness Of Experiments:**

I did not assess the experiments.

**Review Assessment: Thoroughness In Paper Reading:**

I read the paper at least twice and used my best judgement in assessing the paper.

---

> ### Author Response · Authors · 2019-11-06
> **Author response**
>
> Thank you for the review!
>
> > As defined by the paper, fragments are connected subsets derived from relational data.
>
> We would like to clarify that fragments are not necessarily connected.
>
> > The authors derived sets of fragments with constants defined by values in the data and
> > anonymized fragment sets with integer assignments. With potential functions sampled
> > from the anonymized and the true value fragments.
>
> We would like to clarify that we do not sample potential functions. We suspect this might be just a typo in the review but we wanted to clarify this to avoid confusion.
>
> > It is not exactly clear to me why the anonymization of fragments is necessary, but the
> > authors suggest this places a greater focus on the graph structure and minimizes the
> > model acting differently with different constants.
>
> If the potential function can use information about constants then it cannot be used in the inductive setting where the constants in the test data are not the same as constants in the training data. However, it makes sense to use information about constants for the transductive setting (which we do using embeddings that we learn automatically in the NMLN in the transductive setting of knowledge graph completion).
>
>
> > At the same time, however, given the emerging contributions in the area, such as
> > relational neural networks (Poole et al.), although the technical development is sensible,
> > I don’t find the contribution that technically novel, and seems somewhat straightforward.
> > Given the preliminary investigations, this is a reject from my side.
>
> We respectfully disagree on this point.
>
> Indeed, we are aware of the works that use neural networks for relational data, such as RelNN (Kazemi and Poole, AAAI 18) or lifted relational neural networks (Sourek et al, JAIR 2018) and some newer ones as well. While similar from the high level perspective - they all use neural networks on relational data, there is a crucial difference here: to the best of our knowledge, none of these frameworks is capable of learning a joint distributions on relational structures. That is none of these methods combining neural networks and relational learning could be used in the molecule generation experiment. In particular, RelNN is “only” a neural-extension of relational logistic regression. We model the joint probability distributions of the relational structures which is a much more difficult task (also from the computational-complexity point of view).

---

### Official Review · AnonReviewer1 · 2019-10-24
**Official Blind Review #1**

**Rating:** 6

**Review:**

This paper presents Neural Markov Logic Networks  (NMLN), which is a generalization of Markov Logic Networks (MLN). Unlike MLN which relies on pre-specified first-order logic (FOL) rules, NMLN learns potential functions parameterized by neural networks on fragments of the graph. The potential function can possibly take into account the constants present using embeddings to better solve transductive problems（otherwise the potential can only use relational structure). To make computation tractable, the size of local potential functions is constrained. Training of this MRF is performed by solving a min-max entropy problem: conditioned on an informative potential, the uncertainties shall be decreased. Experiments on a knowledge base completion task and a graph generation task show superior performance compared to baselines like neural theorem provers.

Pros:
1. no need to specify FOL rules and can potentially discover subtle relations not evident to us.
2. can be used for generation since the learned rules might be more fine-grained than what we can specify.
3. it's interesting that on nations knowledge base completion problem even without constant embeddings it works fine, which shows the power of just using relational structure.

Cons:
1. the computation complexity of the global potential function grows combinatorally with the clique size k and polynomially with graph size n, which is unrealistic to any larger graphs than the small molecules, if any higher order statistics matters (e.g. in molecules there are rings).

Questions:
1. for training can we use MLE?

Overall this is an interesting work. I think it is a natural generalization of Markov Logic Networks and works on two small problems. I am inclined to recommend this paper to the community.


-----updates after reading rebuttal-----
Thanks for the clarification. I don't have further questions.

**Experience Assessment:**

I have read many papers in this area.

**Review Assessment: Checking Correctness Of Derivations And Theory:**

I carefully checked the derivations and theory.

**Review Assessment: Checking Correctness Of Experiments:**

I carefully checked the experiments.

**Review Assessment: Thoroughness In Paper Reading:**

I read the paper thoroughly.

---

> ### Author Response · Authors · 2019-11-06
> **Author response**
>
> Thank you for the review!
>
> > Cons:
> > 1. the computation complexity of the global potential function grows combinatorally with
> > the clique size k and polynomially with graph size n, which is unrealistic to any larger
> > graphs than the small molecules, if any higher order statistics matters (e.g. in molecules
> > there are rings).
>
> We completely agree and we explicitly admit in the paper as well that scalability is something that we have not solved yet. However, we believe that there are algorithmic ways to improve scalability of the framework for subclasses of NMLNs (first, however, we wanted to present the general framework and show that it can already do interesting things even if only on small relational structures so far). This way the community may also contribute to the development, if people find the framework that we describe here interesting.
>
> For instance, one such subclass could be NMLNs with potentials that are constant for disconnected fragments. Since many real-world domains are sparse, this might turn out to allow us to scale to larger datasets. There are likely other restricted classes of NMLNs that will turn out to be more tractable.
>
> > Questions:
> > 1. for training can we use MLE?
>
> Indeed, solving the min-max entropy problem is shown in the paper to be equivalent to a maximum-likelihood problem. However, the importance of the min-max entropy view is that it (i) dictates the structure of the neural networks and (ii) allows for debugging. Ad (i) the neural networks must have a linear output layer with a learnable weight (this corresponds to the respective lagrange multiplier), otherwise it would not always be possible to guarantee that the NMLN will faithfully model the statistics (potential functions) encoded by the neural network. Ad (ii) We know that the expected value of the potential function encoded by the neural network should be (approximately) the same as the value of the potentials computed on the training instance. This could be used in future to aid the tuning of hyperparameters and for debugging.
>
> > Overall this is an interesting work. I think it is a natural generalization of Markov Logic
> > Networks and works on two small problems.
>
> Thanks, one of the reasons we are excited about this framework is exactly because it is a natural extension of MLNs.

---

### Official Review · AnonReviewer3 · 2019-11-02
**Official Blind Review #3**

**Rating:** 6

**Review:**

I think their strong point is at the same time their weak point. ", they do not rely on explicitly specified first-order logic rules." My question would be, why not use that information if available as prior knowledge? Perhaps I'd like to see a stronger motivation for the use of having to learn this part. I can see how its might be useful but would be very happy to see more motivation for this. I think I've seen Kotler learn the rules of sudoku but sudoku is such a specified problem that Im not convinced yet this is usueful to learn. Thats a different paper but I missed the motivation for that here too.

I like the honesty of the authors for saying it doesn't scale to larger problems. Regardless, I think this paper is good to push the field in that direction. I particularly like the graph generation task. Graph generation, afaik, is not easy.

What I invite the authors to do is to not be restricted by theoretically/principled motivated ways. I believe its better to find things that work well first and then to find a theory (the other way round). This is not enough to reject the paper for me because I do believe this is pushing the field forward in a good direction. If possible I'd suggest to relax the theory and then compare the two models if possible.

In the contribution it says "(i) we introduce a new statistical relational model, which
overcomes actual limitations of both classical and recent related models such as " I would have really liked it to have been spelt out which limitations, very specifically and concisely the paper overcomes in that section.


**Experience Assessment:**

I do not know much about this area.

**Review Assessment: Checking Correctness Of Derivations And Theory:**

I assessed the sensibility of the derivations and theory.

**Review Assessment: Checking Correctness Of Experiments:**

I assessed the sensibility of the experiments.

**Review Assessment: Thoroughness In Paper Reading:**

I made a quick assessment of this paper.

---

> ### Author Response · Authors · 2019-11-06
> **Author response**
>
> Thank you for the review!
>
> > I think their strong point is at the same time their weak point. ", they do not rely on
> > explicitly specified first-order logic rules." My question would be, why not use that
> > information if available as prior knowledge?
>
> In case such information is available, it can be easily added to the NMLN as another potential - one such rule will correspond to one potential function. For instance, let us suppose that we want to add the rule alpha = sm(x) & friends(x,y) => smokes(y). We can add a potential function which counts the number of true groundings of this rule exactly (as in classical MLNs) to the list of potential functions Phi_i with a weight Beta_alpha. We then treat Beta_alpha exactly as the other Beta_i’s and solve the optimization problem.
>
> > In the contribution it says "(i) we introduce a new statistical relational model, which
> > overcomes actual limitations of both classical and recent related models such as " I
> > would have really liked it to have been spelt out which limitations, very specifically and
> > concisely the paper overcomes in that section.
>
> Regarding the recent methods combining neural networks and relational learning, their main limitation compared to NMLNs is that they do not allow expressing joint probability distributions of complete relational structures. On the one hand, it allows them to scale to larger domains for certain problems (e.g. predicting one target predicate from other predicates) but they cannot work for more complex learning and reasoning where one needs to work with the joint probability distribution (one such example is the generation of molecules).
>
> Regarding the more classical methods, such as MLNs (which can model joint probability distributions), their main limitation is the combinatorial nature of the search needed to find the right sets of first-order-logic rules (they also do not allow embeddings which turn out to be useful in the transductive setting).

---

### Decision · Program_Chairs · 2019-12-19

**Decision:**

Reject

**Comment:**

This paper on extending MLNs using NNs is borderline acceptable: one reviewer is strongly opposed, although I confess I don't really understand their response to the rebuttal or see what the issue with novelty is (a position shared by the other reviewers). I'm not sure how to weigh this review, but there is not a lot of signal in favour of rejection aside from the rating.

The remaining two reviews are in favour of acceptance, with their enthusiasm only bounded by the lack of scalability of the method, something they appreciate the authors are upfront about. My view is this paper brings something new to the table which will interest the community, but doesn't oversell the result.

Given the distribution of papers in my area, this one is just a little too borderline to accept, but this is primarily a reflection of the number of high-quality papers reviewed and the limited space of the conference. I have no doubt this paper will be successful at another conference, and it's a bit of a shame we were not in a position to accept it to this one.